



# Potential contributions of nitrifiers and denitrifiers to nitrous oxide sources and sinks in China's estuarine and coastal areas

Xiaofeng Dai[1], Mingming Chen[1], Xianhui Wan[2], Ehui Tan[3], Jialing Zeng[1], Nengwang Chen[1, 4], Shuh-Ji Kao[1, 3], Yao Zhang[1*]

[1]State Key Laboratory of Marine Environmental Science, College of Ocean and Earth Sciences, Xiamen University, Xiamen 361005, China

[2]Department of Geosciences, Princeton University, NJ 08540, USA.

[3]State Key Laboratory of Marine Resource Utilization in South China Sea, Hainan University, Haikou, Hainan, China

[4]Fujian Provincial Key Laboratory for Coastal Ecology and Environmental Studies, College of the Environment and Ecology, Xiamen University, Xiamen 361005, China

*Correspondence to*: Yao Zhang (yaozhang@xmu.edu.cn)

**Abstract**. Nitrous oxide ($N_2O$) is an important ozone-depleting greenhouse gas produced and consumed by microbially mediated nitrification and denitrification pathways. Estuaries are intensive $N_2O$ emission regions in marine ecosystems. However, the potential contributions of nitrifiers and denitrifiers to $N_2O$ sources and sinks in China's estuarine and coastal areas are poorly understood. The abundance and transcription of six key microbial functional genes involved in nitrification and denitrification, as well as the clade II-type *nosZ* gene-bearing community composition of $N_2O$ reducers, were investigated in four estuaries spanning the Chinese coastline. The results showed that the ammonia-oxidizing archaeal *amoA* genes and transcripts were more dominant in the northern Bohai Sea (BS) and Yangtze River estuaries, which had low nitrogen concentrations, while the denitrifier *nirS* genes and transcripts were more dominant in the southern Jiulong River (JRE) and Pearl River estuaries, which had high levels of terrestrial nitrogen input. Notably, the *nosZ* clade II gene was more abundant than the clade I-type throughout the estuaries except for in the JRE and a few sites of the BS, while the opposite transcript distribution pattern was observed in these two estuaries. The gene and transcript distributions were significantly constrained by nitrogen and oxygen concentrations, as well as salinity, temperature, and pH. The *nosZ* clade II gene-bearing community composition along China's coastline had a high diversity and was distinctly different from that in the soil and marine oxygen-minimum-zone waters. By comparing





the gene distribution patterns across the estuaries with the distribution patterns of the $N_2O$ concentration and flux, we found that denitrification may principally control the $N_2O$ emissions pattern.

**1 Introduction**

Nitrous oxide ($N_2O$) is a kind of ozone-depleting substance and an important, long-lived greenhouse gas with 298 times the single mole global warming potential of carbon dioxide ($CO_2$) (IPCC 2007; Ravishankara et al., 2009; Rowley et al., 2013). Prokaryotic microorganisms play an important role in $N_2O$ production and consumption through nitrification and denitrification pathways (Silvennoinen et al.,

2008; Santoro et al., 2011; Babbin et al., 2015; Domeignoz-Horta1 et al., 2015; Ji et al., 2018b; Meinhardt et al., 2018). $N_2O$ is produced as a byproduct in the first step ($NH_4^+ \rightarrow NO_2^-$) of nitrification, which is catalyzed by ammonia monooxygenase in ammonia-oxidizing archaea (AOA) and ammonia-oxidizing bacteria (AOB) (Codispoti and Christensen, 1985). The ammonia monooxygenase subunit A gene (*amoA*) is frequently used as a functional gene marker for AOA and AOB analysis. $N_2O$ is also produced as a

kind of intermediate product in the denitrification process, in which nitrite ($NO_2^-$) is reduced to nitric oxide (NO) and then further reduced to $N_2O$. Usually, the nitrite reductase genes *nirS* and *nirK* are used to evaluate the $N_2O$ production potential through denitrification (Wrage et al., 2001; Shaw et al., 2006; Hallin et al., 2018). Some bacterial nitrifiers can also reduce $NO_2^-$ to $N_2O$ through a nitrifier denitrification pathway. The last step of denitrification is the only known biological $N_2O$ consumption

pathway, reducing $N_2O$ into nitrogen ($N_2$) under the catalysis of nitrous oxide reductase encoded by the *nosZ* gene. This gene is divided into two clades according to the differences in the signal peptides of nitrous oxide reductase (Henry et al., 2006; Jones et al., 2013). Intergenomic comparisons have revealed that approximately 51% of the microorganisms possessing clade II-type *nosZ* genes lack nitrite reductase, do not produce $N_2O$, and thus are expected to drive potential $N_2O$ sinks (Jones et al., 2008; Sanford et

al., 2012; Marchant et al., 2017). The community composition of microorganisms with *nosZ* clade II genes is considered important for the $N_2O:N_2$ end-product ratio of denitrification, influencing the regional $N_2O$ source or sink characteristics (Philippot, 2013; Domeignoz-Horta1 et al., 2015). However, there are few studies on *nosZ* clade II gene diversity and community composition in Chinese estuarine and coastal areas.



Decades of research have revealed that the ocean is the second most important source of $N_2O$ emissions following arable soils, contributing one-third of the $N_2O$ emission fluxes to the atmosphere (Nevison et al., 2003). Estuaries, as important bioreactors, are the most active $N_2O$ exchange areas in the ocean, accounting for 33% of oceanic $N_2O$ emissions with only approximately 0.4% of the area (Bange et al., 1996; Zhang et al., 2010). Denitrification was the major contributor to $N_2O$ production in terrestrial

ecosystems and stream and river networks (Beaulieu et al., 2011; Marzadri et al., 2017). However, complete denitrification can consume $N_2O$ (Jones et al., 2014). A recent study reported a fourfold increase in global riverine $N_2O$ emissions that was influenced by human activities (Yao et al., 2020). Marine nitrification supported by ammonia-oxidizing archaea was largely responsible for oceanic $N_2O$ emissions, especially in the open ocean (Santoro et al., 2011; Löscher et al., 2012), while nitrate reduction

was the dominant $N_2O$ source in oxygen minimum zones (OMZs) (Yamagishi et al., 2007; Ji et al., 2018a). In estuaries, the transition zones between the land and sea, both nitrification and denitrification could be dominant driving processes of active $N_2O$ exchange. For example, nitrification was credited as the dominant $N_2O$ production pathway in the Schelde Estuary as well as in some other European estuaries (De Wilde and De Bie, 2000; Barnes and Upstill-Goddard, 2011; Brase et al., 2017), while an inverse

correlation between $N_2O$ concentration and oxygen indicated that sedimental denitrifiers might be the dominant $N_2O$ contributor in the Potomac River estuary (McElroy et al., 1978). In addition, research in the Chesapeake Bay revealed that physical processes such as wind events and vertical mixing affected the net balance between $N_2O$ production and consumption, resulting in a variable source and sink for $N_2O$ (Laperriere et al., 2019).

The four main estuaries along the Chinese coastline include the Bohai Sea (BS) in the north, the Yangtze River Estuary (YRE) and the adjacent East China Sea (ECS) in the middle, as well as the Jiulong River Estuary (JRE) and Pearl River Estuary (PRE) in the south. The BS is a semi-enclosed sea located in the north temperate zone of China. Influenced by frequent human activities and seasonal variability in inputs from the Yellow River, Liao River, Luan River, and Hai River, seasonal hypoxia is an important

characteristic of the BS (Chen, 2009). The YRE and the adjacent ECS, which receive a large amount of nutrients from the largest river in Asia (Yangtze River: runoff $9.2 \times 10^{11}$ $m^3$ $yr^{-1}$) (Zhang, 2002), also exhibited seasonal hypoxia off the estuary from July to September because of the enhanced primary





productivity (Zhu et al., 2011). Both the JRE and PRE are located in densely populated and industrialized

subtropical areas, with runoffs of $1.44 \times 10^{10}$ m$^3$ yr$^{-1}$ and $3.26 \times 10^{11}$ m$^3$ yr$^{-1}$, respectively (Cao et al., 2005; He et al., 2014). To clarify the potential contributions of nitrification and denitrification to sources and sinks of N$_2$O in China's estuarine and coastal aeras, the abundance and transcription activity of six key microbial functional genes involved in nitrification and denitrification (AOA and AOB *amoA*, *nirS*, *nirK*, *nosZ* clade I and II genes) were investigated in the four estuarine areas. In addition, the *nosZ* clade II gene diversity and N$_2$O reducing community composition were analyzed based on clone libraries to

assess the local N$_2$O sink potential.

**2 Materials and methods**

**2.1 Sampling and biogeochemical parameter measurements**

A total of 228 (130 for DNA and 98 for RNA) samples from fifty-four sites were collected (Fig. 1). One

hundred and sixteen samples (58 for DNA and for 58 for RNA) were collected from 20 stations with two or three depth layers in the BS on the R/V Dongfanghong #2 from August to September 2018. Seventy-four (41 for DNA and 33 for RNA) samples were collected from 16 stations with one to four depth layers in the YRE on the R/V Yanping II from July to August 2017. Water samples were collected using a rosette sampler fitted with Niskin bottles (SBE 911, Sea-Bird Co). Sixteen surface samples (9 for DNA

and 7 for RNA) from a water depth of ~0.5 m were collected from the JRE on the R/V Ocean II during September 2016. Twenty-two samples for DNA were collected from 11 stations with two depth layers in the PRE on the R/V Wanyu during January 2017. Water samples were collected using an organic glass hydrophore (1 L; Kedun Co., China). In addition, 2 and 1 surface sediment samples were acquired using a grab sampler from the JRE in December 2015 and from the YRE from July to August 2017, respectively.

Water samples of 0.2–2.5 L were filtered through 0.22 μm pore size polycarbonate membranes (Millipore, USA) within 1 h at a <0.03 MPa pressure for quantitative PCR (qPCR) analysis. Water samples were serially filtered through 10, 3, and 0.22 μm pore size polycarbonate membranes (Millipore, USA) for clone library analysis (Table S1). The membranes for RNA extraction were immediately fixed with 1.5 mL of RNAlater (Invitrogen, Life Technologies). All filters and sediment samples were quick-

frozen in liquid nitrogen and then stored at –80 °C for laboratory analysis.



Temperature, salinity, and depth  were measured using conductivity temperature depth (CTD) (SBE 911, Sea-Bird Co.) in the BS, YRE, and PRE. In the JRE, water temperature and salinity were continuously measured (every 3 s for 1 min) using a YSI6600D salinometer installed on an underway pumping system (Yan et al., 2019). Dissolved oxygen (DO) concentrations were measured using a WTW

multiparameter portable meter (Multi 3430, Germany). Ammonia was analyzed on deck using the indophenol blue spectrophotometric method. Nitrate, nitrite, and silicate were measured using an AA3 Autoanalyzer (Bran+Luebbe Co., Germany) (Dai et al., 2008).

**2.2 Nucleic acid extraction, clone library, and phylogenetic analysis**

DNA from water samples was extracted using the phenol-chloroform-isoamyl alcohol method (Massana et al., 1997) with minor modifications. DNA from sediment samples was extracted using a FastDNA SPIN Kit for Soil (MP Biomedicals, USA). RNA from water samples was extracted using the RNeasy Mini kit according to the manual (Qiagen, USA). Clean RNA, which was verified by the amplification of the bacterial 16S rRNA gene with the primer set 342F/798R, was reverse transcribed to cDNA by the

SuperScript III first strand synthesis system (Invitrogen, Life Technologies) using random hexamers following the user manual. The quality of both the DNA and cDNA was checked by amplifying the full-length bacterial 16S rRNA gene before storage at –80 °C.

A total of 19 DNA samples (16 from water and 3 from sediment) from the four estuaries (Figs. 1 and 4) were used to construct clone libraries for the clade II-type *nosZ* gene. PCR was run with the primer

set nosZ-II-F (5'-CTIGGICCIYTKCAYAC-3') and nosZ-II-R (5'-GCIGARCARAAITCBGTRC-3') according to a previously reported reaction mixture and program (Jones et al., 2013) with the minor modification of using 10 μg of bovine serum albumin (BSA; Takara, Bio Inc.) instead of T4 gp32. PCR products were purified using an agarose gel DNA purification kit (Takara, Bio Inc.), ligated into the pMD19-T vector (Takara, Bio Inc.), and transformed into high-efficiency competent cells of *Escherichia*

*coli* according to the manufacturer's instructions. Forty to 127 positive *nosZ* clones were randomly selected from each library, reamplified using the vector primers M13-F and RV-M, and sequenced using ABI 3730 automated DNA sequence analyzer (Applied Biosystems). Poor-quality sequences with termination codons were manually checked and removed, and chimeras were removed using UCHIME (Edgar et al., 2011). All sequences were clustered into operational taxonomic units (OTUs) based on a



3% sequence divergence cutoff (Jones et al., 2014; Wittorf et al., 2020). Alpha diversity indices of the

clade II-type *nosZ* gene were calculated using the Usearch package (Edgar et al., 2010). The

representative sequences of OTUs were translated and analyzed with the BLASTp tool (*e*-value $<10^{-5}$).

The top 10 most similar sequences of each OTU were used as references. All sequences were aligned

using MAFFT (Katoh and Standley, 2013) and automatically trimmed using trimAl (Capella-Gutiérrez

et al., 2009). A maximum likelihood (ML) phylogenetic tree was constructed using Fasttree (v2.7.1,

default parameters) (Price et al., 2010) with 500 bootstrap replicates for node support determination.

### 2.3 Quantitative PCR of six functional genes

Archaeal *amoA,* bacterial *amoA*, *nirS*, *nirK*, *nosZ* clade I, and *nosZ* clade II genes were quantified by

qPCR with DNA and cDNA as templates using a CFX96 (Bio-Rad Laboratories, Singapore). Given the

relatively high ammonia concentration in the estuaries, the ammonia-oxidizing archaea (AOA) shallow

cluster (Water Column Cluster A; Francis et al., 2005) was targeted with the primer set Arch-amoAFA

and Arch-amoAR (Beman et al., 2008). Ammonia-oxidizing bacteria (AOB) are mostly affiliated with

two groups: Betaproteobacteria (*β*-AOB) and Gammaproteobacteria (*γ*-AOB) (Lam et al., 2007). Since

the latter was below detection limit in previous studies of Chinese estuaries (Zheng et al., 2017; Hou et

al., 2018), only *β*-AOB was targeted with the primer set amoA-1F and amoA-r New (Rotthauwe and

Witzel, 1997; Hornek et al., 2006). Bacterial *nirS* and *nirK* genes were quantified with the primer sets

nirS-1F and nirS-3R (Braker et al., 1998) and nirK876 and nirK1040 (Henry et al., 2004). Bacterial clade

I-type *nosZ* genes were quantified with the primer set nosZ2F and nosZ2R (Henry et al., 2006).

For the clade II-type *nosZ* gene quantification, the previously published primer sets were found to

have less than 80% amplification efficiency (Jones et al., 2013, 2014; Chee-Sanford et al., 2020). Here,

we designed a new primer set for use in our estuarine samples to quantify this gene. Representative

nucleotide sequences of each OTU obtained from the clone libraries derived from the PRE samples

(n=48) were translated into amino acid sequences and then aligned with the representative reference

sequences (n=116; covering 87 genera) obtained from the Functional Gene Repository

(http://fungene.cme.msu.edu/index.spr) by Clustal W. Two highly conserved regions containing five and

three amino acids in length were chosen to design new primer fragments. The new primer pairs and the

previously published nosZ-II-F and nosZ-II-R primer sets (Jones et al., 2013) were all evaluated by

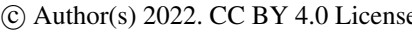

Primer Premier 6.0, and eligible primer sets (GC content: 40–60%; optimal melting temperatures: 52–

58 °C; stable 5' end and specific 3' end with no clamp or complementary structure) were tested by qPCR.

The best primer combination was nosZ-II-F and the newly designed reverse primer (nosZ-II-Rnew:

KGCRTAGTGIGGYTCDCC) with a ~325 bp target fragment length (Fig. S1). The qPCR system is

shown in Table S2, and the optimized qPCR program was as follows: an initial 5 min denaturing step at

95 °C, followed by 35 cycles of 95 °C for 30 s, annealing at 53 °C for 60 s, 72 °C extension for 60 s and

a final extension at 72 °C for 10 min. The coverage of the primer sets was evaluated using the

Search_pcr2 command of Usearch with the 116 reference sequences mentioned above and all clone

sequences (n=1378) obtained from the clone libraries. A coverage of 93.5% (≤2 mismatches) was

obtained for the new primer set.

The presence of PCR inhibitors in DNA extracts was examined by qPCR with different dilutions of

DNA (1-, 10-, and 100-fold dilutions). The samples with inhibitor were diluted 10 times to overcome the

inhibitor effect according to our evaluation. Standard curves were constructed for the six genes using

plasmid DNA from clone libraries generated from the PCR products. qPCRs were performed in triplicate

and analyzed against a range of standards ($10^1$ to $10^8$ copies per μL). All specific primer sequences and

reactions for qPCR/PCR used in this study are shown in Table S2. The amplification efficiencies ranged

from 87% to 109% with $R^2$ >0.99 for each qPCR run. The specificity of qPCR products was verified by

melting curves, agarose gel electrophoresis, and sequencing.

### 2.4 Statistical analysis

Redundancy analysis (RDA) based on qPCR or clone library data was used to analyze variations in the

gene/transcription distribution and *nosZ* clade II community composition under environmental

constraints using R (R Core Team, 2017). The qPCR or clone library-based relative abundances and

environmental factors were normalized via Z transformation (Magalhães et al., 2008). The collinearity

between environmental parameters was excluded (variance inflation factors > 10; Palacin-Lizarbe et al.,

2019). The null hypothesis that the community was independent of environmental parameters was tested

using constrained ordination with a Monte Carlo permutation test (999 permutations). Since a normal

distribution of the individual datasets was not always met, we used the nonparametric Wilcoxon rank-

sum tests for comparing two variables in GraphPad Prism software (San Diego, CA, USA). The bivariate





correlations were described by Spearman's ($\rho$ value) or Pearson's (r value) correlation coefficients. False

discovery rate-based multiple comparison procedures were applied to evaluate the significance of

multiple hypotheses and identify truly significant comparisons (false discovery rate-adjusted $P$ value)

(Pike, 2011).

### 3 Results

### 3.1 Environmental characteristics of the four estuaries

Water temperature increased with decreasing latitude from the BS (16.1–26.4 °C) to the YRE (19.2–

29.1 °C) and JRE (28.7–30.8 °C), where samples were all collected in summer. Samples were collected

in winter in the southernmost PRE, where the water temperature was 19.7–20.5 °C (Fig. 2). Salinity

exhibited consistently high values in all sites of the BS and YRE (26.4–34.6), except for two low values

(14.34 and 21.66) observed in the river mouth. In the JRE and PRE, obvious salinity gradients were

detected from 0.1 to 30.7. The DO concentration varied in the range of 4.25–8.46 mg $L^{-1}$ in the BS, 1.25–

8.71 mg $L^{-1}$ in the YRE, 4.04–6.89 mg $L^{-1}$ in the JRE, and 2.22–9.22 mg $L^{-1}$ in the PRE. There was a

distinct DO gradient from upstream to downstream of the PRE (Fig. 2). The dissolved inorganic nitrogen

(DIN: ammonium, nitrite, and nitrate) concentrations were generally lower in the BS and YRE compared

to those in the JRE and PRE. The ammonium concentration was in the range of 0.006–1.27 μM in the

BS, below detection (BD) to 1.99 μM in the YRE, 7.01–36.78 μM in the JRE, and 1.71–417.38 μM in

the PRE. The nitrite concentration was in the range of BD–5.65 μM in the BS and 0.004–2.5 μM in the

YRE, 7.24–30.87 μM in the JRE, and 0.41–69.23 μM in the PRE. The nitrate concentration ranged from

0.067–13.97 μM in the BS, 0.23–65.09 μM in the YRE, 24.94–241.32 μM in the JRE, and 3.0–320.53

μM in the PRE. Clear DIN concentration gradients were observed from upstream to downstream in the

JRE and PRE, particularly in the PRE.

### 3.2 Distribution of six key functional genes

The abundances of archaeal *amoA*, bacterial *amoA*, *nirS*, *nirK*, *nosZ* I, and *nosZ* II genes showed distinct

distribution patterns among the four estuaries (Figs. 3a–h). We divided the six genes into two groups for

analysis: one group included archaeal and bacterial *amoA*, *nirS*, and *nirK* genes indicating nitrification

and denitrification related to $N_2O$ production (Figs. 3a–d), and the other included bacterial *nosZ* I and

*nosZ* II genes indicating $N_2O$ consumption (Figs. 3e–h). In the gene group of $N_2O$ production-related



processes, archaeal *amoA* was the most dominant in the BS ($2.66\times10^4$–$3.68\times10^8$ copies $L^{-1}$) and YRE

($4.86\times10^3$–$9.47\times10^7$ copies $L^{-1}$) (Wilcoxon test, $P < 0.01$; Figs. 3a, b and Table S3), accounting for 3.96%

to 96.2% and 2.84% to 99.67%, respectively. In contrast to the northern estuaries, archaeal *amoA*

($5.28\times10^5$–$4.40\times10^6$ copies $L^{-1}$) and bacterial *nirS* ($2.57\times10^5$–$6.29\times10^6$ copies $L^{-1}$) genes codominated

the gene group of $N_2O$ production-related processes in the JRE (Fig. 3c), accounting for 2.43% to 72.93%

and 25.03% to 93.77%, respectively. In the southernmost PRE, the *nirS* gene was the most abundant

($3.48\times10^4$–$1.66\times10^9$ copies $L^{-1}$), especially upstream ($P < 0.05$), accounting for 4.24% to 99.91% (Fig.

3d). Generally, archaeal *amoA* was widespread in all samples, and its abundance decreased from north

to south with differences of one to two magnitudes. A similar pattern was observed for bacterial *amoA*,

with lower abundances than archaeal *amoA* (Table S3). The abundance of the *nirS* gene was highest in

the PRE among the four estuaries, while the highest number of copies of the *nirK* gene was present in

the BS (Table S3). Among the different water depths, only the bacterial *amoA* and *nirS* genes in the BS

were observed to be more highly distributed in the middle and bottom layers than in the surface layer by

one to three orders of magnitude ($P < 0.05$).

In the $N_2O$-consuming genes, the abundances of the clade II-type *nosZ* gene were $6.55\times10^3$ to

$2.24\times10^7$ copies $L^{-1}$ in the BS (Fig. 3e), $6.14\times10^3$ to $8.11\times10^6$ copies $L^{-1}$ in the YRE (Fig. 3f), and BD to

$1.17\times10^7$ copies $L^{-1}$ in the PRE (Fig. 3h), outnumbering the clade I-type ($P < 0.01$), with no significant

differences among the three estuaries. However, the clade II-type *nosZ* gene was below the detection

limit in the JRE, and only the clade I-type was detected with a range of $7.15\times10^3$–$2.32\times10^5$ copies $L^{-1}$

(Fig. 3g and Table S3). The numbers of copies of the clade I-type *nosZ* gene were higher in the BS

estuary than in the other three estuaries ($P < 0.01$).

**3.3 Transcription activity of six key functional genes**

For the four genes of $N_2O$ production-related processes, a generally similar relative abundance

distribution pattern was observed between transcripts and genes in the BS (Fig. 3i). Archaeal *amoA* gene

transcripts ($3.51\times10^3$–$1.62\times10^6$ transcripts $L^{-1}$) were significantly more abundant than other transcripts

($P < 0.01$), accounting for 37.94% to 99.30% of the total abundance of gene transcripts (Table S4).

Slightly different from the gene distribution in which the number of copies of the bacterial *amoA* gene

was relatively more abundant than that of the archaeal *amoA* gene in the river mouth of the YRE (Fig.



3b), the archaeal *amoA* gene transcript was abundant in the whole YRE, accounting for 9.1% to 100% of the total abundance of gene transcripts, with a dominant abundance of *nirS* gene transcripts in a few samples (Fig. 3j). A different distribution pattern was also observed between transcripts and genes in the

JRE (Figs. 3c, k). Bacterial *amoA* ($7.06 \times 10^5$–$8.22 \times 10^7$ transcripts L$^{-1}$) rather than archaeal *amoA* transcripts ($P < 0.05$) were codominant with *nirS* transcripts ($5.96 \times 10^5$–$2.31 \times 10^7$ transcripts L$^{-1}$) (Fig. 3k). Notably, the total gene transcript abundance of N$_2$O production-related processes was higher in the JRE ($1.31 \times 10^6$–$9.76 \times 10^7$ transcripts L$^{-1}$) than in the BS and YRE ($3.03 \times 10^2$–$1.12 \times 10^6$ transcripts L$^{-1}$) ($P < 0.01$; Table S4). Bacterial *amoA* gene transcripts, consistent with the gene distribution, significantly

increased with depth in the BS ($P < 0.05$). No significant differences in transcript abundance were observed among different depths for the six functional genes in the YRE.

For the N$_2$O-consuming genes, only the clade I-type *nosZ* gene transcript was determined ($26.2$–$2.34 \times 10^3$ transcripts L$^{-1}$), while the clade II-type *nosZ* gene transcript was below the detection limit in the BS (Fig. 3l; Table S4). However, the *nosZ* II gene transcripts (bellow detection to $1.81 \times 10^5$ transcripts

L$^{-1}$) dominated most stations in the YRE, except for a dominant distribution of the *nosZ* I gene transcript in the river mouth (Fig. 3m). Similar to the gene distribution, in the JRE, only the *nosZ* I gene transcript was determined ($1.23 \times 10^3$–$5.37 \times 10^4$ transcripts L$^{-1}$) (Fig. 3n). No RNA samples were obtained in the PRE.

**3.4 Phylogenetic diversity of the clade II *nosZ* gene**

Clone libraries of *nosZ* clade II were constructed for 19 samples from the four estuaries, resulting in a total of 1378 quality-controlled sequences that were clustered into 441 OTUs at a similarity level of 97%. The sequencing coverage for each clone library ranged from 73.9 to 96.2%. Higher gene diversity of *nosZ* clade II was observed in the water and sediment samples from the JRE and the sediment sample

from the YRE than in the other samples (Fig. S2a). The rarefaction curves of the samples from JRE and the sediment sample from YRE did not reach a plateau (data not shown), suggesting that some of the diversity of *nosZ* clade II remained unsampled. Phylogenetic analysis of the representative sequences of all the OTUs indicated that the clade II *nosZ* gene sequences were grouped with Bacteroidetes, Proteobacteria, Actinobacteria, Chloroflexi, Chlorobi, Ignavibacteriae, Gemmatimonadetes,

Cyanobacteria, and Acidobacteria, in which the OTUs affiliated with Bacteroidetes, Proteobacteria,

Chloroflexi, and Actinobacteria were generally abundant among all samples (Fig. 4a). The OTUs belonging to Bacteroidetes were divided into two clusters according to the topological structure of the phylogenetic tree. One cluster contained the reference sequences mainly from marine habitats and the OTU sequences retrieved from the four estuaries, while the other cluster included the reference sequences

mainly from terrestrial habitats and the OTU sequences retrieved only from the low-latitude subtropical estuaries JRE and PRE. The OTU sequences affiliated with Alpha-, Gamma-, Delta-, Epsilonproteobacteria, and Actinobacteria were retrieved from the four estuaries, and the reference sequences were mainly from marine habitats, while the OTUs related to Betaproteobacteria, Oligoflexia, Chlorobi, and *Candidatus Melainabacteria* were retrieved only from the subtropical estuaries (JRE and

PRE), and the reference sequences were mainly from terrestrial habitats (Fig. 4a). Most known clusters of *nosZ* clade II can be found in our libraries, including a recently identified widespread clade II-type *nosZ* gene affiliated with the class Oligoflexia (Nakai et al., 2014).

A community structure shift of *nosZ* clade II was observed among the four estuaries (Fig. 4b). Bacteroidetes was the most dominant group in the samples from the BS (39.0–68.5%), followed by

Proteobacteria (Gamma-, Delta-, and Alphaproteobacteria; 18.7–26.0%). The sequences phylogenetically grouped into Proteobacteria (Gamma-, Delta-, and Epsilonproteobacteria; 23.0–70.6%) dominated the clone libraries from the YRE, followed by Chloroflexi (6.9–47.3%). The sequences from the JRE were also mainly affiliated with Proteobacteria (Beta-, Gamma-, Delta-, and Alphaproteobacteria and Oligoflexia; 11.8–40.5%), Bacteroidetes (30.9–37.9%), and Chloroflexi (12.1–50.9%). In contrast

to the three estuaries, the sequences affiliated with Bacteroidetes were absolutely dominant in the clone libraries of the PRE (>69.2%). A nonmetric multidimensional scaling (NMDS) analysis indicated that *nosZ* clade II communities from the same estuary were clustered together at a >10% similarity level, except for a  separate cluster of the sediment community from the YRE (Fig. S2b). The *nosZ* clade II communities from the southern estuaries (JRE and PRE) and northern estuaries (YRE and BS) were

clustered separately at a >3% similarity level.

**3.5 Correlations between six key functional genes and environmental factors**

Variations in the gene/transcript distributions under environmental constraints were analyzed by RDA. The first two RDA axes explained 19.98% and 5.36% of the total variation in the gene–environment

relationship (Fig. 5a). Salinity, DO, nitrite, and ammonium concentrations were significantly correlated with gene distribution ($P < 0.01$). The main variation in $N_2O$ source or sink process-related genetic potentials was across a *nirS* vs. archaeal *amoA* abundance gradient. The *nirS*-rich samples corresponded to those from the southern estuaries (JRE and PRE) with higher ammonium and nitrite concentrations. In contrast, the samples with the highest abundance of archaeal *amoA* were located in sites with high

salinity and low ammonium concentrations in the northern estuaries (BS and YRE). Notably, RDA of the gene transcripts and environmental variables clearly separated the transcripts from different estuaries along the axes, which explained 26.4% and 8.27% of the total variation (Fig. 5b). Variation in transcript distribution was significantly correlated with pH, temperature, nitrite, and nitrate concentration ($P < 0.01$). The main variation of these transcripts was distributed across archaeal and bacterial *amoA* vs. *nosZ*

clade II abundance gradients. The archaeal *amoA* transcript-rich samples corresponded to those from the BS and YRE sites with lower temperatures. The bacterial *amoA* gene was actively transcribed in the JRE and positively correlated with nitrite and nitrate concentrations. The *nosZ* clade II transcript-rich samples corresponded to those from the YRE sites with relatively higher pH and temperature. The *nosZ* clade I and *nirS* transcript distributions were also positively correlated with pH and temperature, respectively.

RDA based on the clone library data of the clade II-type *nosZ* gene revealed that the *nosZ* II community composition was significantly affected by temperature ($P < 0.01$; Fig. 5c). The first two RDA axes explained 33.29% and 13.24% of the total variation. The *nosZ* II gene community compositions in the BS may prefer environments with relatively high salinity and temperature. The community compositions in the JRE water may prefer environments with a high temperature (the sediment samples

were not included in this analysis due to a lack of biogeochemical parameters). The *nosZ* clade II microbes in the PRE and YRE may prefer to distribute in environments with high ammonium concentrations.

**4 Discussion**

**4.1 Spatial niche differentiation of functional genes controlled by environmental factors**

There was a distinct large-scale spatial structure among the detected genes, as shown in Fig. 3. Comparing the relative contributions of these functional genes to the total number of gene copies across the study regions, there was a strong negative correlation between the relative abundances of the archaeal



*amoA* gene and bacterial *nirS* gene ($\rho = -0.89$, $P < 0.01$), and they showed contrasting patterns along

salinity and DIN gradients (Fig. S3). Samples from the BS and YRE exhibited high salinity and low DIN

concentrations. The high abundance of the archaeal *amoA* gene in these areas was consistent with

previous findings of nitrifiers comprised predominantly of AOA in estuarine environments with higher

salinity and lower ammonia concentrations because archaeal nitrifiers exhibit a high ammonia affinity

and salinity tolerance (Martens-Habbena et al., 2009; Abell et al., 2010; Bernhard et al., 2010; Zhang et

al., 2014; Hou et al., 2018; Hink et al., 2018; Ma et al., 2019). In contrast, both the JRE and PRE are

typical subtropical eutrophic estuaries with high DIN inputs from surrounding environments (Cao et al.,

2005; Yan et al., 2012b; He et al., 2014). Denitrifying bacteria are more adaptable to environments with

high organic carbon and nitrogen concentrations because they usually have high requirements for

substrates (Braker et al., 2000; Smith et al., 2007; Mosier and Francis, 2010; Wang et al., 2014; Wei et

al., 2015; Lee and Francis, 2017). The presence of nitrogen oxides was also shown to activate *nirK* and

*nirS* gene expression under anoxic conditions (Riya et al., 2017). Thus, the *nirS*-containing group was

more abundant upstream of the JRE and PRE. The significant correlations between DIN and the *nirS*

gene (Fig. S3) and transcript ($\rho = 0.341$, $P < 0.01$; data not shown) were consistent with a previous

conclusion that high anthropogenic N loading stimulates denitrification (Cole and Caraco, 2001; Garnier

et al., 2006; Beaulieu et al., 2011; Yan et al., 2012a).

Previous studies of $N_2O$-consuming gene abundance were mainly focused on terrigenous

ecosystems, e.g., in soil samples, the clade I- and II-type *nosZ* genes ranged from $10^4$ to $10^8$ and $10^4$ to

$10^7$ copies g dry soil$^{-1}$, respectively (Jones et al., 2013, 2014). In marine ecosystems, only the oxygen-

depleted waters and coastal sediments were investigated, where the clade I-type was approximately $10^5$

copies L$^{-1}$ and both clades I and II ranged from $10^5$–$10^7$ copies g wet sediment$^{-1}$, respectively (Wittorf et

al., 2020; Sun et al., 2021). We detected that the number of copies of the *nosZ* gene ranged from $6.59 \times 10^3$

to $2.35 \times 10^8$ copies L$^{-1}$, with an average of $4.94 \times 10^6$ copies L$^{-1}$, in China's estuarine and coastal areas.

There was a strong negative correlation between the relative abundance of the clade I- and II-type *nosZ*

genes ($\rho = -1$, $P < 0.01$), indicating that the two types were affiliated with different groups. The

distribution of *nosZ* (clades I and II) gene transcripts was significantly positively correlated with pH (Fig.

5b), suggesting that acidification of the ocean may accelerate $N_2O$ emissions. $N_2O$ production influenced



by pH has been observed in N-cycling water engineering systems and terrestrial ecosystems (Mørkved et al., 2007; Blum et al., 2018). Therefore, some studies have suggested that liming for acidic soils could mitigate N$_2$O emissions (McMillan et al., 2016; Wang et al., 2017; Senbayram et al., 2019). DO also

shows an important influence on denitrifying genes, which was consistent with a previous conclusion that O$_2$ concentration can impact the expression and metabolism of denitrification genes through protein sensing of oxygen conditions (Qu et al., 2016; Riya et al., 2017). Notably, we found that the distribution and abundance of the *nosZ* gene and the *nirS* or *nirK* genes were distinctly different, indicating that these functional genes were affiliated with different denitrifiers. This may be because not all N$_2$O-consuming

bacteria contain all denitrification genes (Sanford et al., 2012).

**4.2 Gene transcription expression controlled by environmental factors**

The gene transcript abundance showed a certain regional distribution difference with gene abundance (Fig. 3), suggesting that environmental factors might have different influences on gene distribution and

transcript activity. The bacterial *amoA* gene was transcribed actively in the JRE, although the archaeal *amoA* gene prevailed in gene abundance. Frequent water exchange may result in a large amount of the archaeal *amoA* gene from the ocean, but AOB were more active under high ammonium and low salinity conditions. AOB have been indicated to be the primary N$_2$O producer, even in an AOA-dominated environment (Meinhardt et al., 2018). Meta-analysis also revealed that AOB respond more strongly than

AOA to nitrogen addition (Carey et al., 2016). High abundances of bacterial *amoA* and *nirS* gene transcripts make the JRE a more potentially active area of N$_2$O production compared to the northern estuarine and coastal areas, which may be attributed to its high nitrogen input from surrounding environments. In contrast, in the mouth of the YRE, although the bacterial *amoA* gene contributed a large proportion of the gene abundance, the archaeal *amoA* gene was transcribed more actively. Flushing water

from the Yangtze River may transport a large amount of the bacterial *amoA* gene, but the archaeal *amoA* gene was more competitive in low ammonium and oxygen environments (Fig. 2) since the enzyme ammonia monooxygenase in AOA has a higher affinity for ammonia and a lower oxygen requirement than the AOB (Park et al., 2010; Martens-Habbena and Stahl, 2011).

**4.3 N$_2$O emissions potential implied by functional gene distribution**





The community structure of nitrifiers and denitrifiers was thought to have an important influence on $N_2O$ emissions. For example, the abundance and expression of the archaeal *amoA* gene showed comparable patterns with $N_2O$ production in the OMZ of the eastern tropical North Atlantic (Löscher et al., 2012). Inhibition of the abundance of bacterial *amoA* genes in hyperthermophilic composting was proven to decrease $N_2O$ emissions (Cui et al., 2019). The expression of the *nirK* gene induced by the addition of nitrate caused an increase in $N_2O$ production in an anoxic soil slurry experiment (Riya et al., 2017). Transcription of clade I-type *nosZ* mRNA in the lower $N_2O$ emission system was one order of magnitude higher than that in the higher $N_2O$ emission system in wastewater treatment plants (Song et al., 2014). To assess how community structure controls the regional $N_2O$ source or sink potential across China's estuaries, we analyzed the relationships between $N_2O$ concentration, $N_2O$ flux, and $\Delta N_2O$ (data collected from the literature below) and the six functional gene distributions across the four estuaries. The $N_2O$ concentration, $N_2O$ flux, and $\Delta N_2O$ all showed an increasing distribution pattern from the northern, high-latitude to the southern, low-latitude estuaries (Figs. 6a–c), with hot spots in the north and center of the BS, nearshore of the YRE, and upstream of the JRE and PRE (Qinji, 2005; Chen et al., 2008; Zhang et al., 2008; Wu et al., 2013; Song et al., 2015; Wang et al., 2016; Ma et al., 2019; Lin et al., 2020). Notably, total *amoA* gene abundances displayed a contrary pattern, while total *nir* gene abundances and the ratio of total *nir* to *amoA* gene abundances (*nir*/*amoA*) had generally consistent patterns with the $N_2O$ concentration, $N_2O$ flux, and $\Delta N_2O$ across the four estuaries (Figs. 6d–f). A significant correlation was even observed between the $N_2O$ flux and the *nir*/*amoA* ratio based on the four averages of the four estuaries (r = 0.95, n = 4, $P < 0.05$). Therefore, the *nir*/*amoA* ratio can indicate the $N_2O$ emission potential in China's estuaries, which was consistent with previous findings that the $N_2O$ production yield of denitrification was higher than that of nitrification in the lab and in situ experiments (Kester et al., 1997; Löscher et al., 2012; Stieglmeier et al., 2014; Frey et al., 2019).

Notably, the total *nosZ* gene abundance of $N_2O$-reducing denitrifiers seemed to have a contrasting distribution pattern with the $N_2O$ concentration, $N_2O$ flux, and $\Delta N_2O$ across the four estuaries, with higher abundances in the high-latitude BS and lower abundances in the low-latitude JRE (Fig. 6g). The total *nosZ* gene abundances were one to two orders of magnitude lower than the total *nir* gene abundances in the JRE and PRE, where the $N_2O$ concentration and flux were higher than those in the BS and YRE.





This indicated a distinctly higher denitrification-derived $N_2O$ emission potential in the JRE and PRE.

The ratio of total *nir* to *nosZ* clade I gene abundances (*nir*/*nosZ* I) had a highly similar pattern with the $N_2O$ concentration, $N_2O$ flux, and $\Delta N_2O$ across the four estuaries in general (Fig. 6h), and significant correlations were also observed between the $N_2O$ flux and *nir*/*nosZ* I (r = 0.97, n = 4, $P < 0.05$). Therefore, the *nir*/*nosZ* I ratio could be a better indicator of $N_2O$ emission potential in China's estuaries. Abundances and activities of the $N_2O$-producing (*nirS* or *nirK*–bearing) community relative to the $N_2O$-reducing

(*nosZ*-bearing) community have also been used to assess the $N_2O$ emission potential of soils (Thompson, 2016; Zhao et al., 2018). The high load of DIN in estuaries could be responsible for the high denitrification-derived $N_2O$ emission potential. Both the *nir*/*nosZ* and *nir*/*amoA* ratios were positively correlated with the $NH_4^+$, $NO_3^-$, and $NO_2^-$ concentrations (Spearman's $\rho = 0.32$–$0.68$, n = 114–122, $P < 0.01$ for each) and negatively correlated with salinity (Spearman's $\rho = -0.45$—$0.66$, n = 114–122, $P <$

0.01 for each). Previous studies in the YRE have proven that nitrogen input accelerates $N_2O$ production in estuaries (Zhang et al., 2010; Yan et al., 2012a). Therefore, sufficient supplies of substrates may support high rates of denitrification and thus high $N_2O$ emissions.

**4.4 Influence of $N_2O$ emissions by $N_2O$ reducer composition**

The community structure and diversity of the clade II *nosZ* gene retrieved from China's estuaries were different from those previously reported in soil and marine OMZ water (e.g., in the eastern tropical South and North Pacific and Arabian Sea) (Jones et al., 2013, 2014; Sun, 2020). The dominant *nosZ* clade II-bearing groups were affiliated with Bacteroidetes, Chloroflexi, Gamma-, and Betaproteobacteria in our four estuarine and coastal areas. However, the most abundant *nosZ* clade II groups found in the OMZ

were affiliated with *Anaeromyxobacter* (Deltaproteobacteria) and *Marinobacter* (Gammaproteobacteria) (Sun, 2020). The *nosZ* clade II organisms from terrestrial systems showed distinctly higher diversity (Sanford et al., 2012; Jones et al., 2014; Hallin et al., 2018; Zhao et al., 2018; Kato et al., 2018). The phylogenetically distinct predominant $N_2O$ reducers can influence $N_2O$ emissions directly or indirectly (Song et al., 2014). According to genomic information, *nosZ* clade II carriers affiliated with

Deltaproteobacteria and Chlorobi have neither the *nirK* nor *nirS* gene, and less than half of *nosZ* clade II organisms affiliated with Bacteroidetes, Chloroflexi, Gamma-, and Epsilonproteobacteria harbor the *nirK* or *nirS* gene, while all of the *nosZ* clade II microbes affiliated with Alpha- and Betaproteobacteria





also have the *nirS* gene (Hallin et al., 2018). Therefore, the distinct *nosZ* clade II community structure among the four estuaries may contribute to their different N$_2$O emissions potential. For example,

distinctly high diversity of the *nosZ* clade II gene was retrieved from the JRE water and sediment samples as well as the YRE sediment sample compared to the other estuaries. The high diversity of the *nosZ* clade II gene may be caused by the high temperature (e.g., in the low-latitude JRE) and sufficient nutrients at those sites. Previous studies have also indicated that the biodiversity of denitrifying bacteria increased in high-temperature seasons (Castellano-Hinojosa et al., 2017) and that nitrogen availability had a

positive effect on denitrifying bacteria in boreal lakes (Rissanen et al., 2011). In addition, the habitat type may also affect the abundance and diversity of N$_2$O-reducing communities, e.g., silty mud and sandy sediments had higher genetic potentials for N$_2$O reduction than cyanobacterial mat and *Ruppia maritima* meadow sediments (Wittorf et al., 2020).

**5 Summary**

This study revealed the distinct distribution patterns of six key microbial functional genes and transcripts related to N$_2$O production and consumption pathways in the BS, the YRE, the adjacent ECS, the JRE, and the PRE. The archaeal *amoA* genes and transcripts were more abundant in the northern BS, YRE, and the adjacent ECS, while the denitrifier *nirS* genes and transcripts were more abundant in the southern

JRE and PRE. The *nosZ* clade II gene was more abundant than the clade I-type throughout the estuaries except for in the JRE and a few sites of the BS, while the opposite transcript distribution pattern was observed in these two estuaries. Water mass parameters (temperature and salinity), substrates (ammonia/ammonium, nitrite, and nitrate), and influencing parameters of substrate availability (DO and pH) regulated the gene, transcript, and community composition distribution patterns. The community

structure of the clade II-type *nosZ* gene retrieved from China's estuaries was distinctly different from those of the soil and marine OMZ. Furthermore, combined with the N$_2$O concentration, flux, and ΔN$_2$O data collected from previous studies, our analysis found that although both the clade I- and II-type *nosZ* genes of N$_2$O reducers were widely distributed in these estuaries, N$_2$O production by the denitrification pathway may be more important in determining the N$_2$O emissions patterns across the estuaries. Nitrogen

loads may influence the N$_2$O source and sink processes by regulating the distribution of the related functional microbial groups.



**Data availability**

All quality-controlled sequences were submitted to GenBank with accession numbers OM567739–

OM568649. All other data can be accessed in the form of Excel spreadsheets via the corresponding

author.

**Supplement**

The Supplement related to this article is available online.


**Author contributions**

YZ conceived and designed the study. XD, XW, MC, ET, and NC performed the experiments and

auxiliary data collection. XD analyzed the data. XD and YZ wrote the paper. All authors contributed to

the interpretation of the results and critical revision.


**Competing interests**

The authors declare no conflicts of interest.

**Acknowledgments**

We thank Zuhui Zuo, Yufang Li, and Minyuan Liu for their assistance in sampling and DNA/RNA

extraction, as well as Jiaming Shen for his valuable comments and suggestions  in the preparation of the

manuscript. Thanks are also given to CEES Open Cruise for the Jiulong River Estuary - Xiamen Bay and

Shuiying Huang and Jiezhong Wu for their organizational help.

**Financial support**

This research was supported by the NSFC projects (42125603, 41721005, 92051114, and  42188102).

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






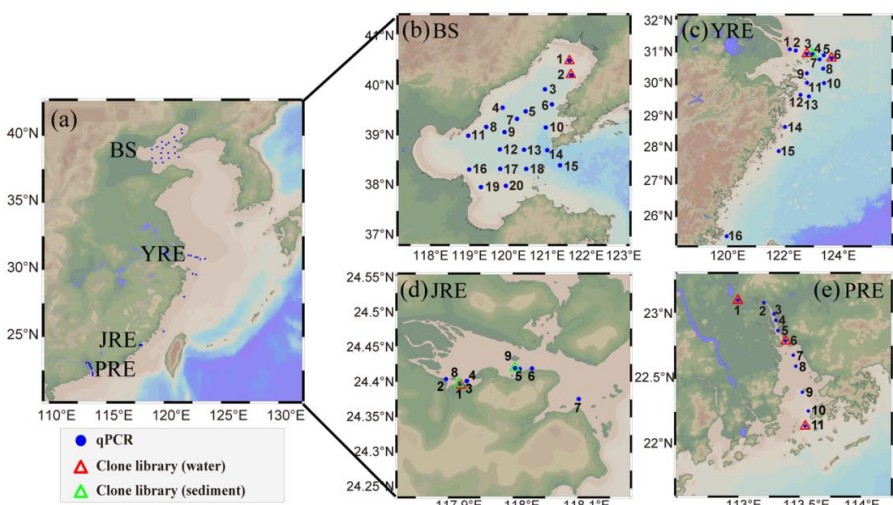

**Figure 1. (a)** Sampling sites in the four estuaries along China's coastline; **(b)** Bohai Sea (BS); **(c)** Yangtze River Estuary (YRE); **(d)** Jiulong River Estuary (JRE); **(e)** Pearl River Estuary (PRE). The figure was produced by Ocean Data View 5.2.0 (http://odv.awi.de/).




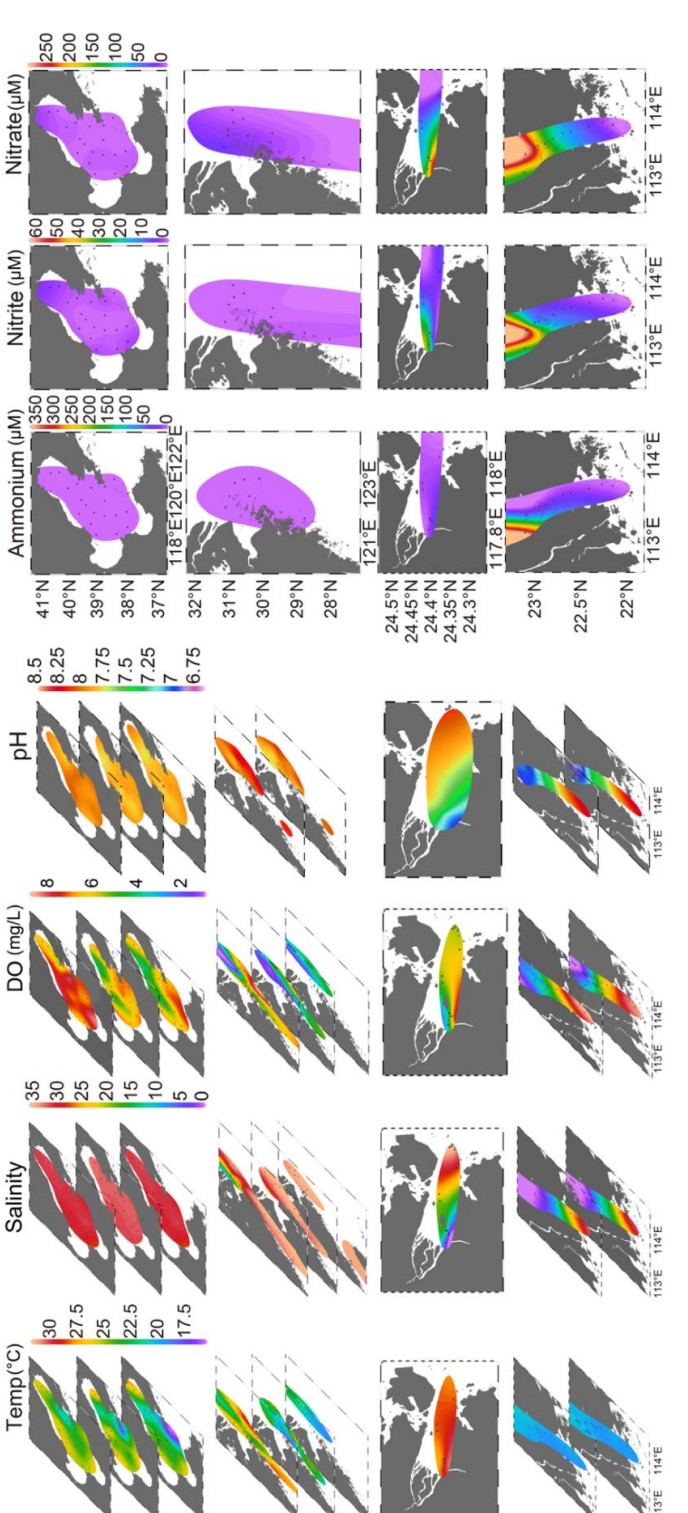

**Figure 2.** Temperature (Temp), salinity, dissolved oxygen (DO), pH, ammonium, nitrite, and nitrate concentration distributions in the four estuaries. Depth integrated mean values were used for ammonium, nitrite, and nitrate concentration distributions.

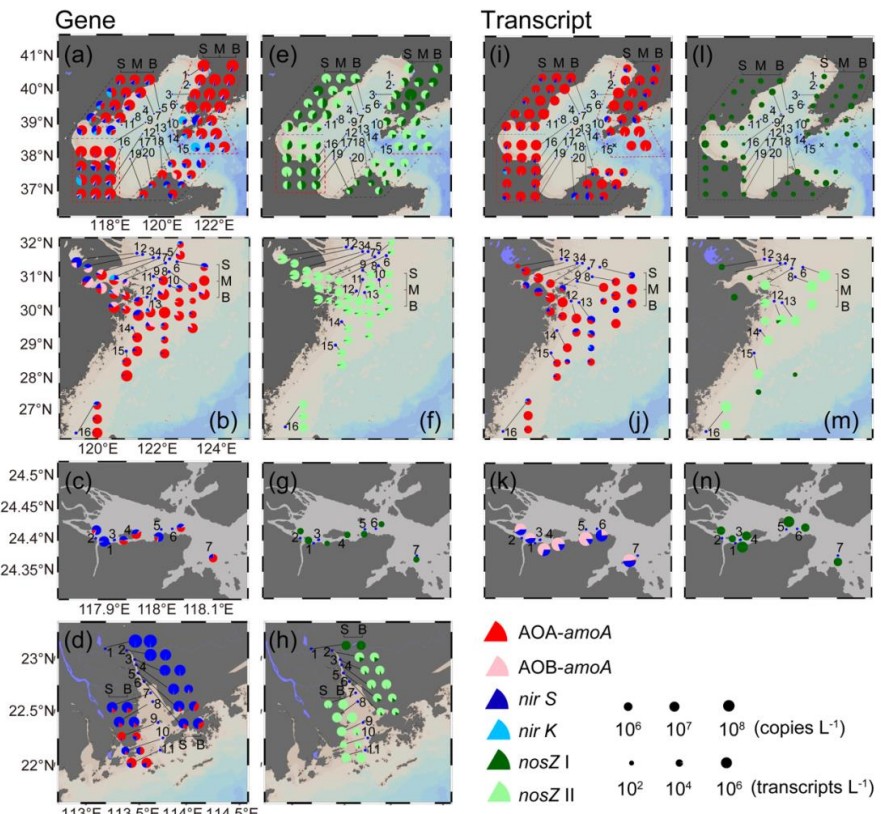

**Figure 3.** Six key functional gene and transcript abundance distributions in the four estuaries. S, surface layer; M, middle layer; B, bottom layer. **(a)**–**(d)** Gene related to $N_2O$ production; **(e)**–**(h)** Gene related to $N_2O$ consumption; **(i)**–**(k)** Transcript related to $N_2O$ production; **(l)**–**(n)** Transcript related to $N_2O$ consumption. **(a)**, **(e)**, **(i)**, and **(l)** Bohai Sea; **(b)**, **(f)**, **(j)**, and **(m)** Yangtze River Estuary; **(c)**, **(g)**, **(k)**, and **(n)** Jiulong River Estuary; **(d)** and **(h)** Pearl River Estuary.



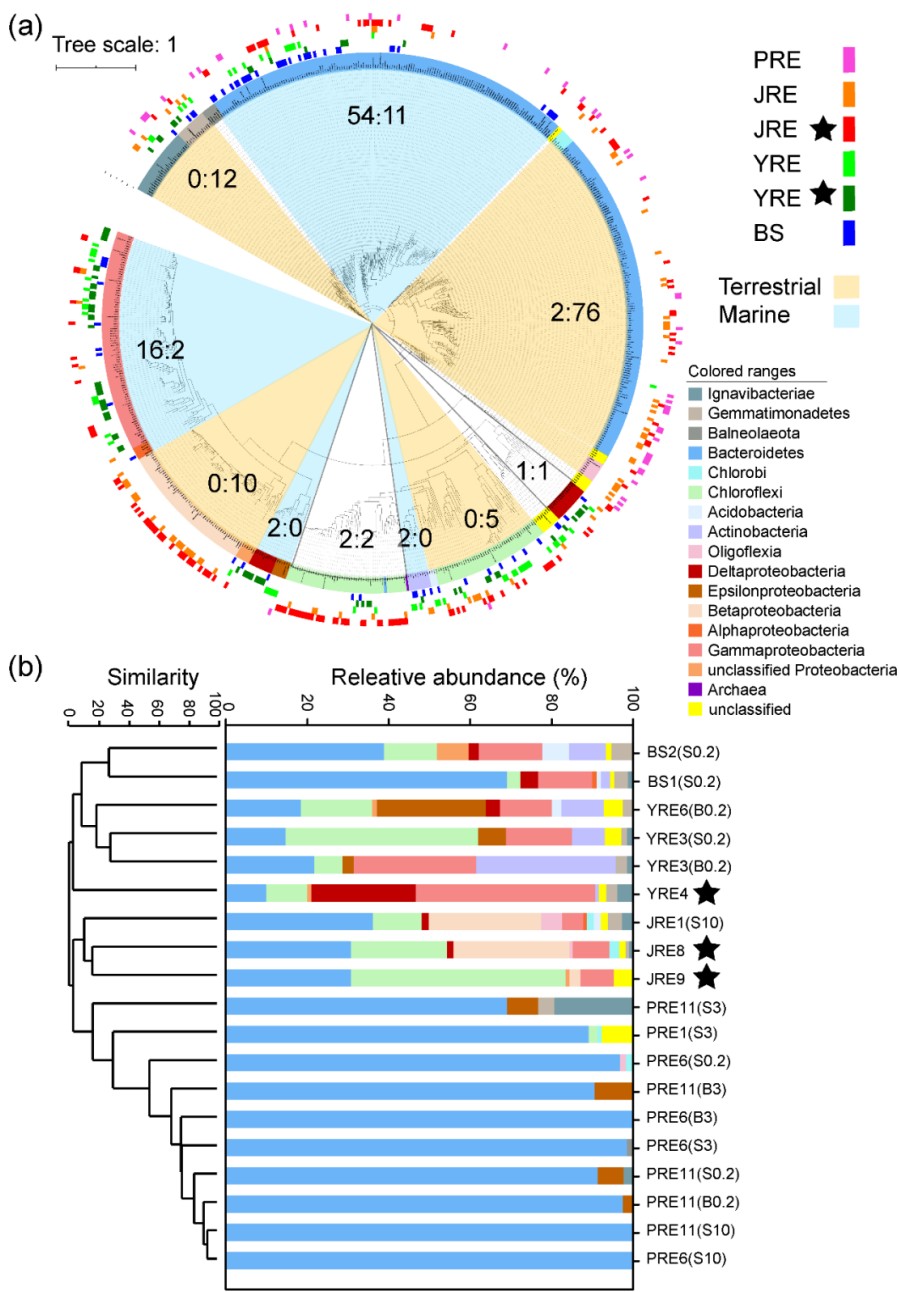

**Figure 4. (a)** Maximum likelihood phylogenetic tree of amino acid sequences of the clade II-type *nosZ*. The colors of the inner circle indicate taxonomic affiliations based on reference sequences. The colors of the outer circles represent the sources of clone sequences. The phylogenetic tree was bootstrapped 500 times. The scale bar represents the number of amino acid substitutions per site. Numbers before and after the colons indicate the number of reference sequences from marine and terrestrial habitats, respectively. The figure was produced using the interactive tree of



life (http://itol.embl.de/; Letunic and Bork 2016). **(b)** Relative abundances of community compositions of the clade II-type *nosZ* gene clone libraries in the four estuaries. The colors of the bars indicate taxonomic affiliations. The similarity was calculated from Bray–Curtis similarity. Black stars indicate sediment samples.

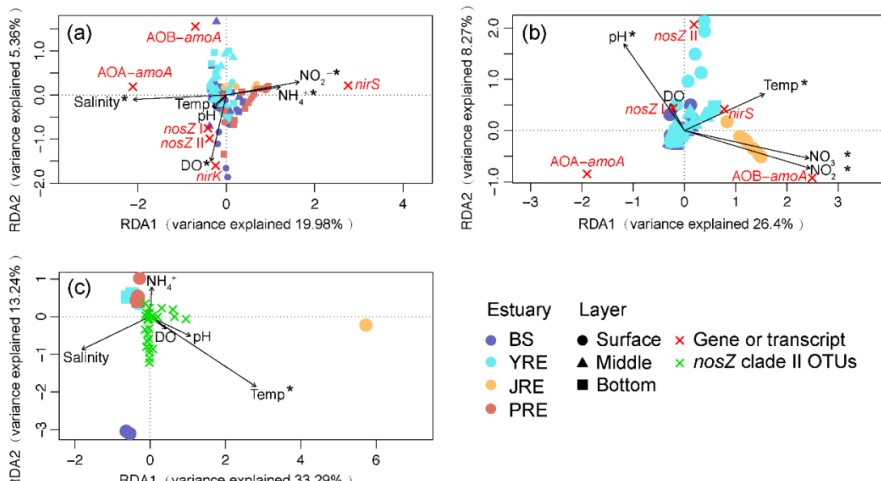

**Figure 5.** Redundancy analysis of the relative abundances of ammonia-oxidizing archaeal *amoA* (AOA-*amoA*), bacterial *amoA* (AOB-*amoA*), *nirS*, *nirK*, and *nosZ* clade I and II **(a)** genes and **(b)** transcripts, as well as of **(c)** the community composition of the *nosZ* clade II clone libraries under biogeochemical constraints. Each circle, triangle, or square represents an individual sample from the surface, middle, or bottom layer, respectively. The fork-shaped symbol represents the functional gene, transcript, or *nosZ* clade II OTU. Vectors represent environmental variables. Asterisks indicate statistically significant variables. Temp, temperature; DO, dissolved oxygen.





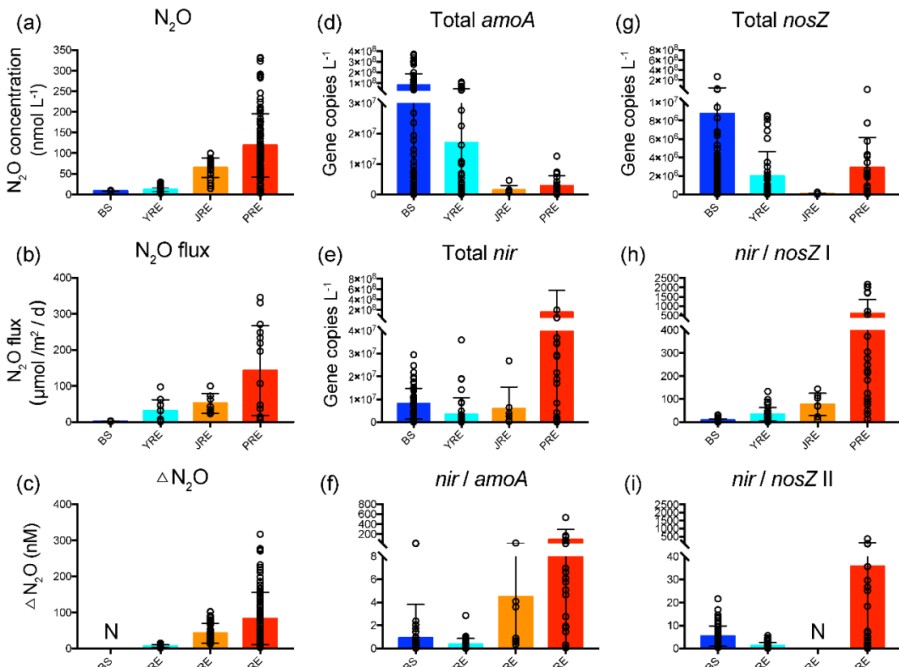

**Figure 6.** The ranges of **(a)** N₂O concentration, **(b)** N₂O flux, **(c)** ΔN₂O (data from Qinji, 2005; Chen et al., 2008; Zhang et al., 2008; Wu et al., 2013; Song et al., 2015; Wang et al., 2016; Ma et al., 2019; Lin et al., 2020), **(d)** total archaeal and bacterial *amoA* gene abundance, **(e)** total *nirS* and *nirK* gene abundance, **(f)** the ratio of total *nir* to *amoA* gene abundance, **(g)** total *nosZ* clade I and II gene abundance, **(h)** the ratio of total *nir* to *nosZ* clade I gene abundance, and **(i)** ratio of total *nir* to *nosZ* clade II gene abundance in the Bohai Sea (BS), Yangtze River estuary (YRE), Jiulong River estuary (JRE), and Pearl River estuary (PRE). Black circles represent the value of each sample. Bars represent the mean values. Error bars indicate standard deviation. N, no data or not determined.