# Peer review of "Potential contributions of nitrifiers and denitrifiers to nitrous oxide sources and sinks in China's estuarine and coastal areas"

_Biogeosciences, 2022_

## Author Response (AR1)

**Response to editor**

**Comments to the author**:

Dear Dr. Zhang,

Thank you for submitting your manuscript "Potential contributions of nitrifiers and denitrifiers to nitrous oxide sources and sinks in China's estuarine and coastal areas" to Biogeosciences. Two expert reviewers have assessed your submission and highlighted that the work provides new insight into estuarine microbial $N_2O$ cycling and the diversity of nitrogen cycling microbial populations.

I appreciate your thoughtful and thorough responses to the reviewers during the interactive peer review phase. At this time, I invite you to revise your paper considering all of the reviewers' comments, and resubmit it for further review. When submitting the revised version please include a copy showing all changes marked to assist in my final review of the manuscript.

Feel free to contact me if you have any questions.

Best,
Denise

Thank you very much for handing and assessing our manuscript. We have revised the manuscript based on the comments/suggestions. The point-by-point replies to the comments are in blue color as below.

**Response to reviews #1**

Dai et al. investigated the distribution of six key genes and transcripts related to $N_2O$ production and consumption in four main estuaries in China. The authors analyzed the correlation between these genes and $N_2O$ fluxes or concentrations obtained from previous literature and discussed what environmental factors might control the gene distribution pattern. This work implies denitrification might be essential for $N_2O$ emissions in these estuaries. This study provides new insight into microbial divers of $N_2O$ cycling in estuaries in China.

**Response:**

Thank you very much for taking the time to review our manuscript. The point-by-point reply to the comments are in blue color as below.

Here are some minor suggestions:

Line 70: cite (Frey et al., 2020 Biogeosciences) for the dominance of nitrate derived $N_2O$ in OMZs.

**Response:**

Thanks for your suggestion. We have added the citation of the article in the revised manuscript. (Line 74)

Lines 75-79: cite (Ji et al. 2018 Biogeosciences) for $N_2O$ production from denitrification in the Chesapeake Bay.

**Response:**

Thank you. We have added the citation in the revised manuscript.

"*In addition, the incubation experiments with nitrogen stable isotope tracer reveal active $N_2O$ production by denitrification in the Chesapeake Bay (Ji et al., 2018b). Another research in the Chesapeake Bay reveals that physical processes such as wind events and vertical mixing affected the net balance between $N_2O$ production and consumption, resulting in a variable source and sink for $N_2O$ (Laperriere et al., 2019).*" (Lines 80-84)

Line 82: cite Figure 1 for the location of the four estuaries.

**Response:**

Thank you. Figure 1 has been cited in the revised manuscript. (Line 87)

Line 126: What were the minor modifications? Please elaborate.

**Response:**

Thank you. The modifications have been elaborated in the revised manuscript.

"*DNA from water samples was extracted using the phenol-chloroform-isoamyl alcohol method (Massana et al., 1997) with minor modifications to maximize the DNA output. Briefly, tubes containing shredded filters, approximately 0.5 g of 0.1 mm glass beads, and 800 µL of STE lysis buffer (0.75 M sucrose, 50 mM Tris-HCl, 40 mM EDTA) were first agitated for 60 s on a FastPrep machine (MP Biomedicals, Solon, OH, USA) at 4.5 m $s^{-1}$. Then, the mixture was processed with lysozyme (1 mg $ml^{-1}$), proteinase K (0.5 mg $ml^{-1}$), and sodium dodecyl sulfate (SDS) (1%) sequentially. At last, the lysate was extracted twice with phenol-chloroform-isoamyl alcohol and once with chloroform-*

*isoamyl alcohol. DNA was precipitated with isopropyl alcohol and washed with 75% ethyl alcohol before dissolved in 50 μL sterile water.*" (Lines 131-139)

Line 145: What kind of alpha diversity? (e.g. Shannon alpha diversity?)

**Response:**

We have revised it as "*Alpha diversity indices (Shannon, Simpson, and Chao1) of the clade II-type nosZ gene were calculated ...*". (Lines 160-161)

Line 148: 'The top 10 most similar sequences of each OTU were used as references.' It is not clear how the taxonomy of the OTU was assigned. Did you use the taxonomy of the top 1 reference as the taxonomy of the OTU or the dominant taxonomy among all 10 references? Please explain this.

**Response:**

Thank you. These reference sequences were first deduplicated, then the representative sequences of OTUs along with these deduplicated reference sequences were used to build a maximum likelihood phylogenetic tree, and the taxonomy of the OTU was assigned according to the structure of the phylogenetic tree. The relevant statements have been added in the revised manuscript for clarification.

"*The top 10 most similar sequences of each OTU were used as references. The deduplicated reference sequences and the representative sequences of OTUs were aligned using MAFFT (Katoh and Standley, 2013) and automatically trimmed using trimAl (Capella-Gutiérrez et al., 2009). A maximum likelihood (ML) phylogenetic tree was constructed using Fasttree (v2.7.1, default parameters) (Price et al., 2010) with 500 bootstrap replicates for node support determination. The taxonomy of the OTU was assigned according to the phylogenetic relationship.*" (Lines 163-168)

Line 235: accounting for % and % of $N_2O$ production-related gene abundance

**Response:**

Thank you. We have revised it as suggested. (Line 252)

Line 240: I believe you meant to say 'one to two orders of magnitudes'.

**Response:**

Thank you. We have corrected it. (Line 258)

Line 291: should be (Figure 4b). Abundance was not reflected in Figure 4a.

**Response:**

Corrected. (Line 308)

Line 376: need to tune down this sentence here since $N_2O$ emission is controlled by both $N_2O$ production and consumption. You could say 'suggesting that acidification of the ocean may decrease $N_2O$ consumption potential.'

**Response:**

Thank you. We have revised it as suggested. (Line 397)

Line 345: Since the four estuaries were sampled in different seasons, it would be useful to see some discussion about how different seasons might affect the distribution of genes and transcripts.

**Response:**

Thank you for the comment. The discussion about the effect of different seasons on the distribution of genes has been added in the revised manuscript.

"*There was a distinct large-scale spatial structure among the detected genes, as shown in Fig. 3. The different sampling seasons between the PRE (January) and the other three estuaries (June to September) may influence the spatial distribution of functional genes across the four estuaries. However, the niche differentiation of functional genes, spatially or temporally, is controlled by environmental factors in essence, such as temperature, salinity, oxygen and nutrient availabilities, and primary productivity.*" (Lines 363-367)

Lines 388-404: *nosZ* clade I was transcribed more even though *nosZ* clade II genes were more abundant (Figure 3 i). The discrepancy between *nosZ* DNA and transcripts is worth discussing.

**Response:**

Thank you. The discussion about the discrepancy between the *nosZ* DNA and transcripts has been added in the revised manuscript.

"*The nosZ clade I gene was transcribed more actively even though the nosZ clade II gene was more abundant (e.g., the case in the BS shown in Fig. 3e and l). The higher growth yields of clade II-type $N_2O$-reducing bacteria than those of clade I-type (Yoon et al., 2016) may lead to a preponderance of the nosZ clade II gene. However, a*

*microbial culture of clade I-type N₂O-reducing bacteria has been reported to have the capability of continually synthesizing N₂O reductase enzyme under oxic conditions to allow for a rapid transition into anoxic environments (Lycus et al., 2018). Such a strategy could result in the more abundant nosZ clade I transcripts observed in the estuaries.*" (Lines 428-435)

Lines 415-416: (a) are these datasets measured from the same months or years as the microbial samples? Or they are mean values of some sort? Please provide a little more detail here. (b) why use gene abundance as indicators but not transcripts? The latter shows 'activity' in some sense. Could you present the transcript data in Figure 6 or the supplement?

**Response:**

Thank you very much for the comments.

(a) The datasets contain almost all available published data on N₂O concentration, N₂O flux, and ΔN₂O in the four estuaries, covering January to November from 2002 to 2015. We suppose that these data could have well covered the annual variations of these parameters in the estuaries for multiple years. The detailed statements have been added in the revised manuscript. A supplementary Table S5 has also been added to the supplementary materials.

"*To assess how community structure controls the regional N₂O source or sink potential across China's estuaries, we collected the data on N₂O concentration, N₂O flux, and ΔN₂O in the four estuaries from the literature, covering January to November from 2002 to 2015 (Table S5; Chen et al., 2008; Lin et al., 2016, 2020; Ma et al., 2019; Song et al., 2015; Wang et al., 2014, 2016; Wu et al., 2013; Xu et al., 2005; Zhan et al., 2011; Zhang et al., 2008, 2010), and analyzed their relationships with the six functional gene distributions.*" (Lines 446-451)

(b) According to your suggestion, a supplementary Fig. S4 presenting the transcript data (see below) has been added to the supplementary materials and the citation of Fig. S4 has been added in the revised manuscript. —"*Similarly, the functional gene transcript distribution indicated that the nir/nosZ I and nir/amoA gene transcript abundance ratios also had consistent patterns with the N₂O concentration, N₂O flux, and ΔN₂O across the four estuaries in general (Fig. S4).*" (Lines 474-476)

Given the transcript datasets contain much fewer data points compared with the gene datasets due to lacking samples from the PRE and the *nirK* transcript data from the JRE

as well as some data below the detection limit, we only present the gene data in Fig. 6 of the main text.

[Figure]

**Fig. S4.** The ranges of (a) N₂O concentration, (b) N₂O flux, (c) ΔN₂O, (d) total archaeal and bacterial *amoA* gene transcript abundance, (e) total *nirS* and *nirK* gene transcript abundance, (f) the ratio of total *nir* to *amoA* gene transcript abundance, (g) total *nosZ* clade I and II gene transcript abundance, (h) the ratio of total *nir* to *nosZ* clade I gene transcript abundance, and (i) ratio of total *nir* to *nosZ* clade II gene transcript abundance in the Bohai Sea (BS), Yangtze River estuary (YRE), Jiulong River estuary (JRE), and Pearl River estuary (PRE). Black circles represent the value of each sample. Bars represent the mean values. Error bars indicate standard deviation. N, no data or not determined.

Line 456: additional citations should be included here: (Bertagnolli et al., 2020 Environmental Microbiology reports) and (Sun et al., 2017 Frontiers in Microbiology).
**Response:**
Thank you. The citations have been added.
"*However, the most abundant nosZ clade II groups found in the OMZs of the eastern tropical South and North Pacific and the Arabian Sea are affiliated with Anaeromyxobacter (Deltaproteobacteria) (Sun et al., 2017; Sun et al., 2021) and those*

*in the coastal OMZ waters of the Golfo Dulce, Costa Rica are affiliated with Gammaproteobacteria, Marinimicrobia, Bacteroidetes, and SAR324 (Bertagnolli et al., 2020).*" (Lines 488-492)

Figure 1: I suggest adding sampling time for each estuary in the figure.

**Response:**

Thank you. Added. (Line 870)

Figure 2: please label the four estuaries (maybe as row names for all subplots). Latitudes and longitudes for the first few plots were missing. Please add latitudes and longitudes for all subplots. It is hard to tell ammonia, nitrite, and nitrate concentrations in three out of the four estuaries. You could use a different scale bar for PRE, so the other three plots could have a better resolution.

**Response:**

Thank you very much for the suggestions. Figure 2 has been modified. Latitudes and longitudes for all subplots have been added, and different scale bars have also been added to tell ammonia, nitrite, and nitrate concentrations in the four estuaries clearly. (Line 874)

Figure S2: red and orange in the plots were too similar to each other, please choose another color to distinguish the two better.

**Response:**

Thank you! We have changed the color and it is better now.

References:

Bertagnolli, A. D., Konstantinidis, K. T. and Stewart, F. J.: Non-denitrifier nitrous oxide reductases dominate marine biomes, Environ. Microbiol. Rep., 12(6), 681–692, doi:10.1111/1758-2229.12879, 2020.

Blackmer AM, Bremner JM.: Inhibitory effect of nitrate on reduction of $N_2O$ to $N_2$ by soil microorganisms, Soil Biol Biochem., 10(3):187–191, doi:10.1016/0038-0717(78)90095-0, 1978.

Levipan, H. A., V. Molina, and C. Fernandez.: Nitrospina-like bacteria are the main drivers of nitrite oxidation in the seasonal upwelling area of the Eastern South Pacific

(Central Chile ~36°S), Environ Microbiol Rep., 6:565–73, doi:10.1111/1758-2229.12158, 2014.

Lycus P, Soriano-Laguna MJ, Kjos M, Richardson DJ, Gates AJ, Milligan DA, et al.: A bet-hedging strategy for denitrifying bacteria curtails their release of $N_2O$, Proc Natl Acad Sci USA, 2018;115:11820–5, doi:10.1073/pnas.1805000115, 2018.

Molina, V., Belmar, L., and Ulloa, O.: High diversity of ammonia-oxidizing archaea in permanent and seasonal oxygen-deficient waters of the Eastern South Pacific, Environ. Microbiol., 12: 2450–2465, doi:10.1111/1462-2920.14246, 2010.

Sobarzo, M., Bravo, L., Donoso, D., Garcés-Vargas, J., and Schneider, W.: Coastal upwelling and seasonal cycles that influence the water column over the continental shelf off central Chile. Prog Oceanogr 75: 363–382., doi:10.1016/j.pocean. 2007. 08. 022, 2007.

Sun, X., Jayakumar, A. and Ward, B. B.: Community composition of nitrous oxide consuming bacteria in the oxygen minimum zone of the Eastern Tropical South Pacific, Front. Microbiol., 8(JUN), 1–11, doi:10.3389/fmicb.2017.01183, 2017.

**Response to reviews #2**

I have carefully read the manuscript "Potential contributions of nitrifiers and denitrifiers to nitrous oxide sources and sinks in China's estuarine and coastal areas" by Dai et al. The manuscript describes the spatial distribution and abundance of marker genes and transcripts related to the production and consumption of nitrous oxide in four estuaries in coastal China. Moreover, the diversity of the clade II-type *nosZ* gene was further investigated along the same four estuaries. The manuscript is nicely written and provides valuable information on the potential mechanisms controlling nitrous oxide consumption/production in coastal China. Furthermore, the results are put in context by using water column physicochemical data and by previously published measurements of nitrous oxide fluxes in the same four estuaries. I have, however, the following minor comments on this manuscript:

**Response:**

Thank you very much for taking the time to review our manuscript. The point-by-point reply to the comments are in blue color as below.

Line 39: add "the" before nitrification.

**Response:**

Thank you. We have added it. (Line 39)

In the introduction, between lines 52-59 the authors describe the physiology/ecology of microorganism possessing the clade II-type *nosZ*. How is that different from microbes containing the clade I-type?

**Response:**

Thank you for the comment. We have added the relevant statements on the calde I-type *nosZ* in the revised manuscript.

"*The microorganisms possessing clade I-type nosZ genes are mainly affiliated with alpha-, beta-, and gamma-proteobacteria, and the clade I gene has a higher frequency of co-occurrence with nir and nor genes than the clade II gene. The nosZ clade II genes are present in a much larger range of archaeal and bacterial phyla (Jones et al., 2013), and intergenomic comparisons have revealed that more than half of the microorganisms possessing clade II genes lack nitrite reductase or nitric oxide reductase, do not produce N₂O, and thus are expected to drive potential N₂O sinks (Graf et al., 2014; Jones et al., 2008; Marchant et al., 2017; Sanford et al., 2012).*" (Lines 52-59)

Check verb tense in the introduction. Usually, present tense is used.

**Response:**

Thank you! We have carefully checked the verb tense and revised many of them into the present tense in the introduction. (Lines 64, 68, 70, 71, 72, 73, 79, 82, 83, and 92)

In material and methods please provide the depths from which samples were collected.

**Response:**

The depth ranges from which samples were collected were provided in the revised manuscript. The depth for each sample can be found in supplementary Table S1. (Lines 106, 108, and 112)

Lines 125-126. What minor modifications?

**Response:**

Thank you. The modifications have been elaborated in the revised manuscript.

*"DNA from water samples was extracted using the phenol-chloroform-isoamyl alcohol method (Massana et al., 1997) with minor modifications to maximize the DNA output. Briefly, tubes containing shredded filters, approximately 0.5 g of 0.1 mm glass beads, and 800 µL of STE lysis buffer (0.75 M sucrose, 50 mM Tris-HCl, 40 mM EDTA) were first agitated for 60 s on a FastPrep machine (MP Biomedicals, Solon, OH, USA) at 4.5 m s$^{-1}$. Then, the mixture was processed with lysozyme (1 mg ml$^{-1}$), proteinase K (0.5 mg ml$^{-1}$), and sodium dodecyl sulfate (SDS) (1%) sequentially. At last, the lysate was extracted twice with phenol-chloroform-isoamyl alcohol and once with chloroform-isoamyl alcohol. DNA was precipitated with isopropyl alcohol and washed with 75% ethyl alcohol before dissolved in 50 µL sterile water."* (Lines 131-139)

Was the same qPCR program (lines 178-180) used for all primer sets?

**Response:**

We are sorry for the unclear statements. Different qPCR programs were used for each primer set and they were mentioned in supplementary Table S2. We have revised the relevant statement in the revised manuscript. —*"All specific primer sequences, reactions, and programs for qPCR/PCR used in this study are shown in Table S2."* (Lines 205-206)

Among the six functional genes, only the qPCR program for the primer set designed by this study for the clade II-type *nosZ* gene quantification was shown in the main text (Lines 195-197).

Line 199: What characteristic of the community? (i.e., community assembly or structure?)

**Response:**

Thank you! We have revised "community" as "*community structure*". (Line 216)

Please add units to salinity values (ppt, I guess?)

**Response:**

Yes, the unit to salinity is ppt or ‰. We have added. (Lines 230 and 231)

When reporting the qPCR/RT-qPCR copy numbers, it is nice that the authors provided the range for each site. However, the median could also be informative, since the extremes may be outliers.

**Response:**

Thank you for the suggestion. The medians of the qPCR/RT-qPCR copy numbers for each site have been added in supplementary Table S3. Basically, nearly no outliers were used.

Line 241: Orders of magnitude?

**Response:**

Thank you. Corrected. (Line 258)

Line 283: It was not clear to me what did the authors mean by "sequencing coverage"?

**Response:**

Sorry for the unclear statement. We have revised "The sequencing coverage for each clone library …" as "the coverage of each lone library ….". We also have added the calculation equation in the Methods section —"*The coverage (C) of each clone libarary was calculated by C = 100% [1 – (n / N)] (Mullins et al., 1995), where n is the number of unique OTUs and N the total number of clones in a library.*" (Lines 158-160)

Line 311-313. NMDS is only a visualization approach, I think the authors measured the similarity level by performing pairwise comparisons of the Bray Curtis dissimilarity index.

**Response:**

Thank you. The descriptions have been modified in the revised manuscript.

"*...at a >10% Bray-Curtis similarity level*" (Line 329)

"*...at a >3% Bray-Curtis similarity level*" (Line 332)

In the discussion, as well as in the introduction, when describing previous literature, the present tense is usually preferred.

**Response:**

Thank you! We have carefully checked the verb tense and revised many of them into the present tense throughout the introduction and discussion sections. (Lines 372, 384, 387, 390, 399, 405, 417, 418, 426, 438, 439, 459, 485, 493, 505, 506, and 508)

Lines 374-376: Authors mention that the *nosZ* gene/transcript abundance was correlated with pH. Could it also be a confounding effect of DIN concentration, since

pH seems to have a similar spatial gradient as DIN, with higher pH and lower DIN in the open ocean (Fig. 2).

**Response:**

Thank you for the comment. When redundancy analysis (RDA) was performed, the collinearity between environmental parameters had been excluded (variance inflation factors > 10; Palacin-Lizarbe et al., 2019). It means that in theory, the correlation observed between gene/transcript abundance and pH was not caused by collinearity between pH and DIN concentration. However, indeed the *nosZ* gene/transcript abundance also correlated with DIN according to Fig. 5a and b. We have added the relevant discussion in the revised manuscript.

*"The nosZ genes and transcripts showed significantly negative correlations with nitrate and/or nitrite (Fig. 5a and b), and similar correlations were also found in mountain lake habitats (Palacin-Lizarbe et al., 2019). It is possible that high abundances of nosZ gene and transcript lead to high consumption of nitrate and nitrite. In addition, it was reported that the presence of nitrate can inhibit the reduction of $N_2O$ to $N_2$ (Blackmer and Bremner, 1978)."* (Lines 400-405)

As I mentioned above, it is a strength of the manuscript to use $N_2O$ flux, and delta$N_2O$ data to put in context the results. However, greater background/information on where, when, and how that data was collected would be helpful to the reader.

**Response:**

Thank you very much for the suggestion! A supplementary Table S5 has been added to the supplementary materials, providing information on where, when, and how the data was collected. The relevant statements have also been added to the main text.

"*To assess how community structure controls the regional $N_2O$ source or sink potential across China's estuaries, we collected the data on $N_2O$ concentration, $N_2O$ flux, and $\Delta N_2O$ in the four estuaries from the literature, covering January to November from 2002 to 2015 (Table S5; Chen et al., 2008; Lin et al., 2016, 2020; Ma et al., 2019; Song et al., 2015; Wang et al., 2014, 2016; Wu et al., 2013; Xu et al., 2005; Zhan et al., 2011; Zhang et al., 2008, 2010), and analyzed their relationships with the six functional gene distributions.*" (Lines 446-451)

As reviewer 1 mentioned, I also wonder why the authors decided to use the gene abundance, and not the transcript abundance to correlate it with $N_2O$ flux and delta$N_2O$.

**Response:**

Thank you! According to your suggestion, a supplementary Fig. S4 presenting the transcript data (see below) has been added to the supplementary materials and the citation of Fig. S4 has been added in the revised manuscript. —"*Similarly, the functional gene transcript distribution indicated that the nir/nosZ I and nir/amoA gene transcript abundance ratios also had consistent patterns with the $N_2O$ concentration, $N_2O$ flux, and $\Delta N_2O$ across the four estuaries in general (Fig. S4).*" (Lines 474-476) Given the transcript datasets contain fewer data points compared with the gene datasets due to lacking samples from the PRE and the *nirK* transcript data from the JRE as well as some data below the detection limit, we only present the gene data in Fig. 6 of the main text.

[Figure]

**Fig. S4.** The ranges of (a) $N_2O$ concentration, (b) $N_2O$ flux, (c) $\Delta N_2O$, (d) total archaeal and bacterial *amoA* gene transcript abundance, (e) total *nirS* and *nirK* gene transcript abundance, (f) the ratio of total *nir* to *amoA* gene transcript abundance, (g) total *nosZ* clade I and II gene transcript abundance, (h) the ratio of total *nir* to *nosZ* clade I gene transcript abundance, and (i) ratio of total *nir* to *nosZ* clade II gene transcript abundance in the Bohai Sea (BS), Yangtze River estuary (YRE), Jiulong River estuary (JRE), and Pearl River estuary (PRE). Black circles represent the value of each sample. Bars

represent the mean values. Error bars indicate standard deviation. N, no data or not determined.

Figure 2: What were the depth layers? Adding a label explaining which panel corresponds to which estuary may be helpful for the reader (same for Fig 3).

**Response:**

Thank you for the suggestion! We have added the labels of estuaries and depth layers in Fig. 2 and Fig. 3. The figure legends have also been modified. (Lines 874 and 875)

Figure 6: Could it help to log transform the qPCR/RT-qPCR data to plot it in order to avoid breaking the y-axis?

**Response:**

Thank you! We have tried to log transform the qPCR/RT-qPCR data to plot it (see below). It seems that using raw data can show the differences between different estuaries better and can be compared clearly with the distribution of $N_2O$ concentration and flux across the estuaries. So we still kept the original version of Figure 6.

[Figure]

Fig. R2-1. The ranges of log-transformed (a) total archaeal and bacterial *amoA* gene abundance, (b) total *nirS* and *nirK* gene abundance, (c) total *nosZ* clade I and II gene abundance in the Bohai Sea (BS), Yangtze River estuary (YRE), Jiulong River estuary (JRE), and Pearl River estuary (PRE). Black circles represent the value of each sample. Bars represent the mean values. Error bars indicate standard deviation.

References:

Graf, D.R.; Jones, C.M.; Hallin, S.: Intergenomic comparisons highlight modularity of the denitrification pathway and underpin the importance of community structure for $N_2O$ emissions, PLoS ONE, doi:10.1371/journal. pone. 0114118, 2014.

Lin, Hua, Dai, Minhan, Kao, Shuh-Ji, Wang, Lifang, Roberts, Elliott, Yang, Jin-Yu Terence, Huang, Tao, He, Biyan: Spatiotemporal variability of nitrous oxide in a large

eutrophic estuarine system: The Pearl River Estuary, China, Marine Chemistry, doi: 10.1016/j.marchem.2016.03.005, 2016.

Zhan, Liyang., Chen, Liqi., Zhang, Jiexia., and Zheng, Airong.: Distribution of $N_2O$ in the Jiulongjiang River Estuary and estimation of its air-sea flux during winter, Journal of Oceanography in Taiwan Strait, 30(MAY), doi:10.3969/J ISSN.2011.02.006, 2011.